# Multipurpose chemical liquid sensing applications by microwave approach

Ayşegül Karatepe[1], Oğuzhan Akgöl[1], Yadgar I. Abdulkarim[2,3], Şekip Dalgac[1], Fahmi F. Muhammadsharif[4], Halgurd N. Awl[5], Lianwen Deng[2], Emin Ünal[1], Muharrem Karaaslan[1], Luo Heng[2], Shengxiang Huang[2]*

1 Department of Electrical and Electronics, Iskenderun Technical University, Hatay, Turkey, 2 School of Physics and Electronics, Central South University, Changsha, Hunan, China, 3 Physics Department, College of Science, University of Sulaimani, Sulaimani, Iraq, 4 Department of Physics, Faculty of Science and Health, Koya University, Koya, Iraq, 5 Department of Communication, Engineering College, Sulaimani Polytechnic University, Sulaimani, Iraq

* hsx351@csu.edu.cn

## Abstract

In this work, a novel sensor based on printed circuit board (PCB) microstrip rectangular patch antenna is proposed to detect different ratios of ethanol alcohol in wines and isopropyl alcohol in disinfectants. The proposed sensor was designed by finite integration technique (FIT) based high-frequency electromagnetic solver (CST) and was fabricated by Proto Mat E33 machine. To implement the numerical investigations, dielectric properties of the samples were first measured by a dielectric probe kit then uploaded into the simulation program. Results showed a linear shifting in the resonant frequency of the sensor when the dielectric constant of the samples were changed due to different concentrations of ethanol alcohol and isopropyl alcohol. A good agreement was observed between the calculated and measured results, emphasizing the usability of dielectric behavior as an input sensing agent. It was concluded that the proposed sensor is viable for multipurpose chemical sensing applications.

## Introduction

Microstrip patch antennas are the most widely used antennas owing to their geometric structure, lightweight, cost effectiveness and easy applicability. The main reason for rapid development of the microstrip patch antennas can be due to the innovations that are brought about by the non-electrical properties of the antenna structure. Its low profile and lightweight making it easily adapt to the microwave integrated circuits.

Nowadays, microstrip patch antennas have found themselves in different application areas such as satellites [1], telecommunications [2], wearable electronic applications [3], imaging devices and sensors [4, 5]. Various approaches were considered to increase the detection, accuracy and gain of the sensors for the detection of chemical liquids [6, 7]. Alongside the development and accelerated sensing technology, the scientists and engineers are studying sensors to be used in a more sensitive way for different fields of applications. The use of antennas for the

**Data Availability Statement:** All relevant data are within the manuscript.

**Funding:** This work was supported in part by the National Key Research and Development Program

of China (Grant no. 2017YFA0204600), the National Natural Science Foundation of China (Grant no. 51802352), teaching reform for postgraduate students of Central South University (Grant no. 2019JG085) and the Fundamental Research Funds of the Central South University (Grant no. 2018zzts355) to SH.

**Competing interests:** The authors have declared that no competing interests exist.

determination of dielectric properties of liquids was first reported by Mirshekar-Syahkal in 1999 [8].

A review of literature revealed that the use of sensors to determine the dielectric properties of liquid materials is realizable. Researchers have successfully employed the sensors to estimate the distance between antennas as well to determine the dielectric parameters of liquid materials [9, 10]. The fabrication of chemical sensors has also been realized using microporous polymer thin films [11]. In another study, Liu *et. al.* [12] determined the dielectric response of small liquids by using metamaterial-based sensor. This sensor was produced by the microstrip feed method resonating at 1.9 GHz, which was used to accurately differentiate ethanol and methanol from the water. Furthermore, carbon/polyurethane dielectric nanocomposite was developed for the applications of capacitive strain sensor [13].

A multi-sensing application was also followed by Altintas *et. al.* to sense the density, rotation, and voltage with the help of metamaterial-based microwave sensors [14], whereby a precise determination of ethanol alcohol ratio was achieved in the frequency range from 3 to 5 GHz. Gregory and Clarke utilized RF and microwave frequencies for measuring the complex permeability of polar liquids [15]. Furthermore, Ebrahimi *et. al.* used microstrip line-based chemical sensors to determine the ratio of water-ethanol composition [16]. It is emphasized that chemical materials based sensors used for the detection of ethanol, methanol and acetone can be configured for various multipurpose sensing applications [17]. Along this line, metamaterials-based signal absorbers were employed to determine the amount of ethanol in the frequency range from 4.42 to 3.97 GHz, with the absorption of up to 90% [18]. Eight-mode antennas were also considered as chemical sensors for ethanol detection thanks to the finding of meaningful correlation between resonant frequency and ethanol concentration [19]. In 2011, Pal *et. al.* designed a new micro resonator antenna structure with the potential to detect and identify small-sized samples of biomaterials. It has been proposed that the sensing effect can be determined by the change in reflection of the antenna due to the varied dielectric constant of the detected material [20]. One of the uses of biosensors is the measurement of ethanol in wine production. Ethanol has been widely used in medicine, biotechnology and food industry for many years. When the ethanol concentration reaches toxic levels during fermentation and distillation, it causes infection of the nasal mucosa, conjunctiva and skin irritation. In addition, alcohol intoxication may occur at higher ethanol concentration levels. Therefore, ethanol analysis is of great importance. Ethanol is a polar liquid which is highly sensitive to the change of temperature. The relaxation frequency of ethanol solution at lowest concentration of 0.9% was found to be 17.83 GHz [21]. Many analytical methods have been developed for the determination of ethanol and other aliphatic alcohols [21]. However, these methods are costly, require long analysis time to implement and are very complex.

Another alternative way to accurately and quickly identify ethanol is to use PCB-based microstrip-fed antennas. In wine production, monitoring of glucose and ethanol during the fermentation process is important to control the quality, specific taste and flavor of the wine [22]. Nowadays, sensor applications have become the matter of interest in both academia and industry. Scientists are working on sensor design to solve various problems.

In the current research work, two practical real-time and high-precision multi-purpose liquid sensors, based on printed circuit board (PCB) microstrip rectangular patch antenna, are proposed and fabricated in order to be used for a multi-task sensing of liquid mixtures. The designed sensors detect the proportions of ethanol alcohol in wine content and isopropyl alcohol content in disinfectant content. The novelty of this work is to suggest a simple and cheap patch antenna sensor which can be readily utilized for the determination of ethanol ratio in wine and disinfectant. Additionally, the new proposed designs can be used to develop a portable sensor instrument to be easily worked with in remote areas.

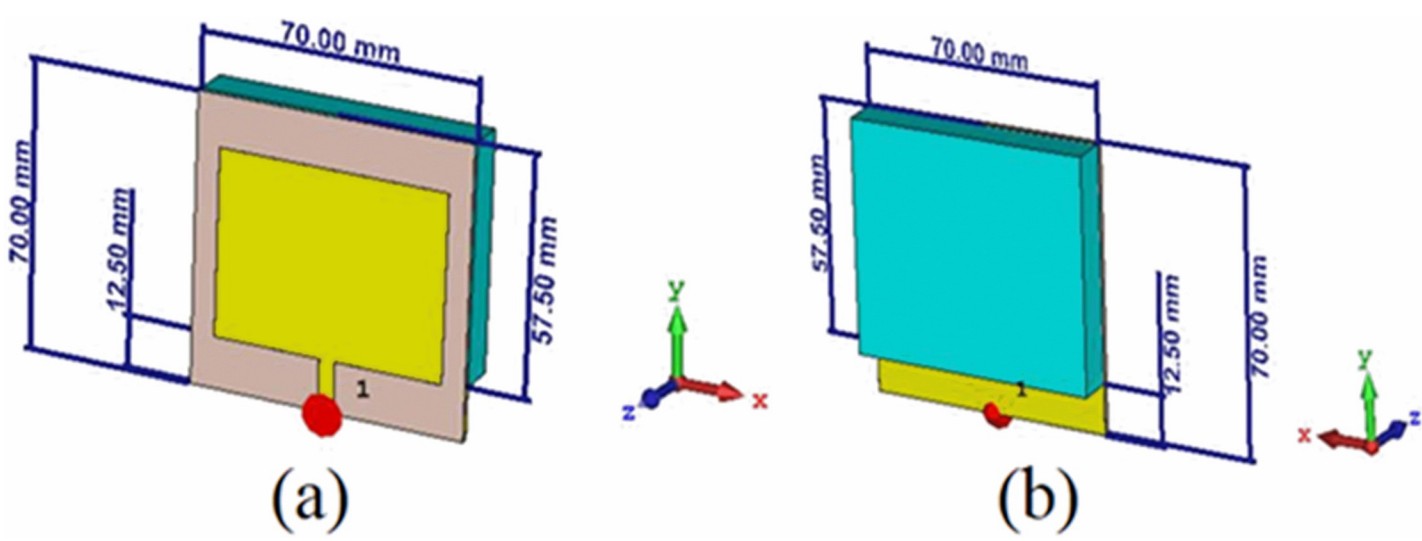

**Fig 1.** Dimension of the designed PCB rectangular patch antenna: (a) front view and (b) backside view.

## Materials and methods

The proposed multi-purpose liquid sensor is based on PCB rectangular patch antenna, as shown in Fig 1. Finite Integration Technique (FIT) based high-frequency electromagnetic solver, CST microwave studio was used to design the sensor structure. In the numerical simulation, Flame Retardant 4 (FR4) substrate was utilized because of its low loss, high mechanical strength, low cost and easy availability. The thickness of the FR4 material was 1.6 mm with a dielectric constant of 4.2, magnetic permeability of 1 and loss tangent value of 0.02. The front face of the resonator was made of copper, with thickness of 0.035 mm and electrical conductivity of $5.8001 \times 10^7$ S/m. The geometry of the PCB-based rectangular patch antenna was determined by means of parametric sturdies in numerical optimizations.

During the analysis of the antenna performance, a discrete port was defined in the antenna structure. The radiating patch portion of the antenna is rectangular and the dimensions of the optimized system were determined to be 45.00 mm and 58.10 mm. The microstrip line used for feeding is at the midpoint of the 58.10 mm latitude and 4 mm in thickness. A slot of 57.5 mm long and 70 mm wide was opened at the backside of the antenna in order to accommodate for the sample holder.

The proposed structure was realized by designing a Printed Circuit Board (PCB) based rectangular patch antenna. In the first step, the dielectric constant and loss tangent values of various sample contents were measured in the frequency range from 1 to 5 GHz in order to be utilized as input variable.

The difference in dielectric properties of the materials under detection is used as a sensing agent. Hence, for the sensing action to take place by the proposed microstrip patch antenna, it is not necessary to have measurement of the dielectric parameters. However, recording the dielectric constant values of the studied liquids is to explain sensing mechanism of the proposed sensor. Therefore, it would be hard for the conventional sensors to detect the materials whose dielectric properties are close enough to each other. Consequently, the PCB based patch antenna sensor is developed in order to increase the sensitivity of detection. The PCB patch antenna sensor was examined and validated based on the measurement results of two different proportions of isopropyl alcohol and ethanol alcohol in disinfectant. Dielectric constant of the samples with different ratios were measured by a KEYSIGHT Network Analyzer (model:

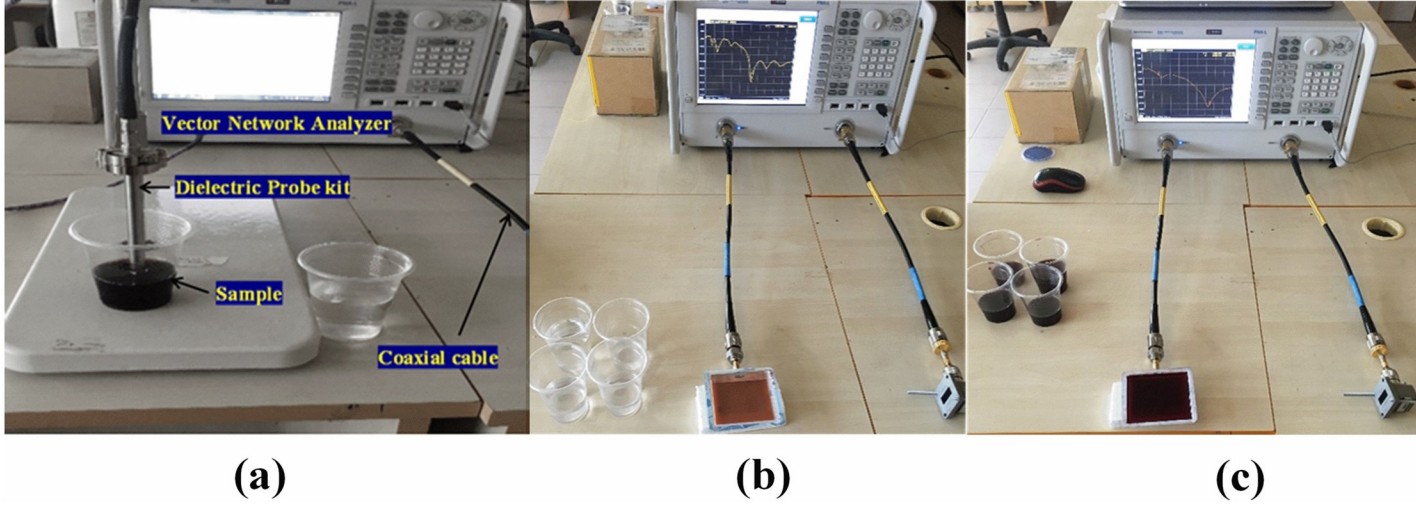

**Fig 2.** The photo of (a) experimental setup to measure the dielectric parameters, (b) VNA connected PCB rectangular patch antenna for disinfectant-Isopropyl measurement and (c) for wine–ethanol sample measurement.

PNA-L N5234A) and a dielectric probe device. Fig 2(A) shows the dielectric measurement setup for the samples using vector network analyzer (VNA). The analyzer was calibrated before taking the measurements. The intended frequency range in the measurements was assigned to be from 1 to 5 GHz. In the first step of calibration, the value of dielectric parameters of water sample at room temperature (25°C) was given to the analyzer.

The air is then measured while the dielectric probe is idle. The dielectric probe was utilized to measure dielectric parameters of the samples at the specified operating frequency from 1 to 5 GHz. In the next step, the probe was immersed in water and the device was calibrated accordingly. Afterward, the calibration apparatus was installed and the dielectric constant of water was measured in order to ensure the presence of a correct calibration for the device. In this way, the values of the measured dielectric parameters and dielectric loss were imported into the simulation program.

## Results and discussion

Table 1 shows the dielectric constants and loss tangent values obtained for different contents of wine-ethanol ratio. The initial proposition of the alcohol in wine was 15%. The dielectric measurement was carried out at room temperature, where the real dielectric value ($\varepsilon$'), the imaginary part ($\varepsilon$'') and the loss tangent (tan $\delta = \varepsilon$''/$\varepsilon$') were found for different ratios. Results showed that the dielectric constant of the liquid samples is highly dependent on the ethanol content. The increment in ethanol ratio has led to decrease in the real part of the dielectric parameters (see Table 1).

**Table 1. Dielectric parameters and loss tangent value for the wine-ethanol samples.**

| Ethanol ratio in wine (%) | $\varepsilon$' | $\varepsilon$'' | Tan $\delta$ |
|---|---|---|---|
| 15 | 63.18 | 23.84 | 0.377 |
| 25 | 52.33 | 24.96 | 0.476 |
| 35 | 43.97 | 24.63 | 0.560 |
| 100 | 6.81 | 6.19 | 0.90 |

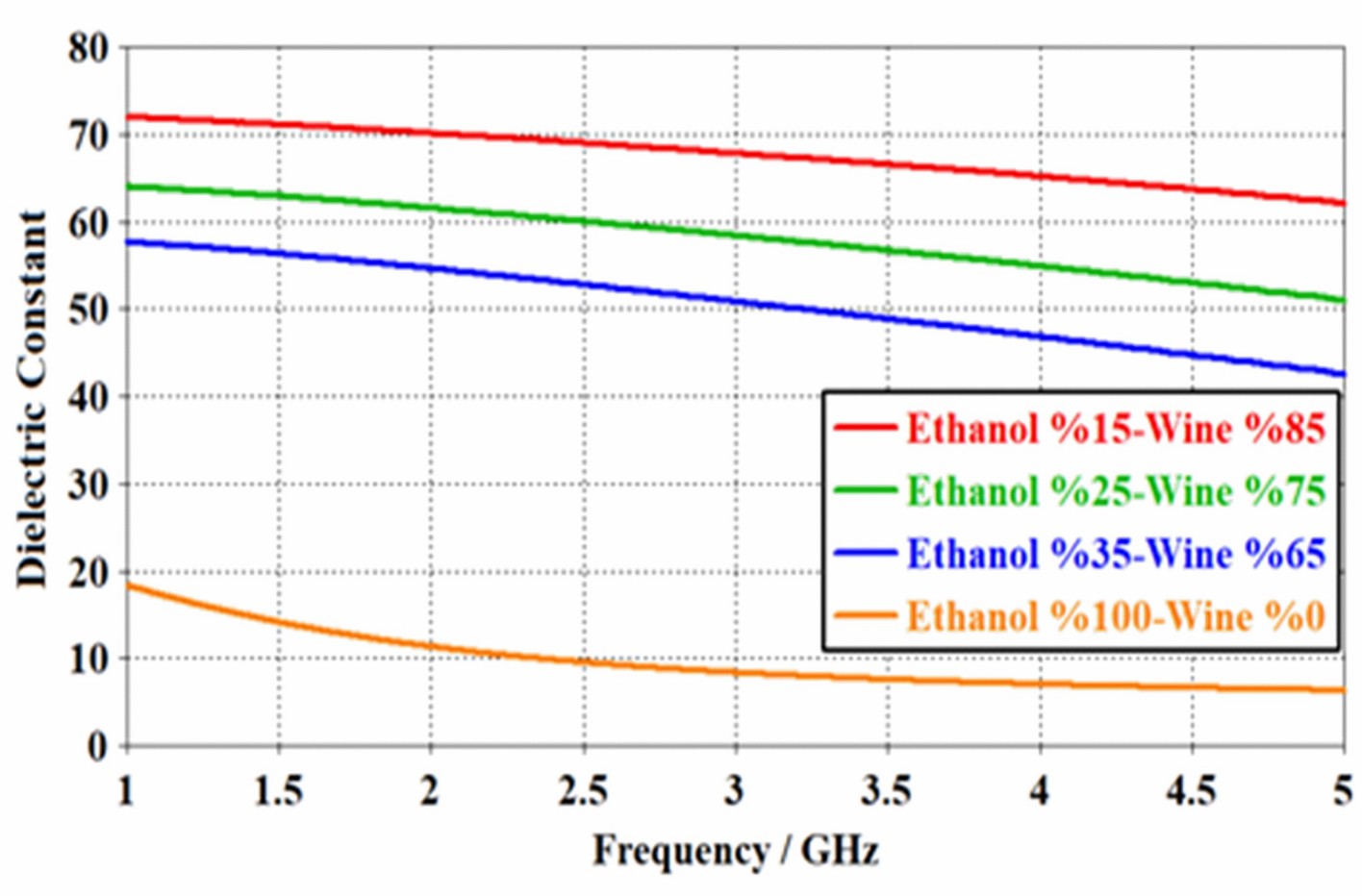

**Fig 3. Dielectric values for different contents of wine-ethanol in the frequency range from 1 to 5 GHz.**

One can notice that the dielectric constant of the wine samples with 15%, 25% and %35 of ethanol ratio presented a real dielectric value of about 63.18, 52.33 and 43.97 at the frequency of 4.656 GHz, respectively. This ratio is equivalent to 6.81% for pure ethanol solution. The pure wine contains 15% of ethanol. This percentage might be changed based on the wine brand. Hence, the wine with 15% of ethanol can be comparable to the pure wine (100%). It was seen that when the amount of ethanol in the sample content has increased, a linear decrease in the real value ($\varepsilon'$) and loss tangent (tan $\delta$) value was obtained. This is ascribed to the expression of the electromagnetic wave energy loss during the transmission, which is increased linearly with the increase of the ethanol ratio in the wine. In addition to that, the more ethanol content in the solution sample has resulted in the decreased dielectric constant in the frequency range from 1 to 5 GHz, as shown in Fig 3. This can be attributed to the week dipole moment of the ethanol molecules compared to those of the wine.

Table 2 shows the measured real dielectric constant ($\varepsilon'$), imaginary dielectric constant ($\varepsilon''$) and the loss tangent (tan $\delta$ = $\varepsilon''/\varepsilon'$) for different ratio of isopropyl alcohol in the disinfectant. One can see from the table that at a fixed frequency of 4.656 GHz, the values of dielectric parameters are decreased with the increase of isopropyl content in the liquid mixtures of isopropyl disinfectant. Noticeably, the real dielectric constant of 70%, 80% and 90% isopropyl were found to be 13.33, 9.59 and 7.37, respectively.

**Table 2. Dielectric parameters and loss tangent values for the disinfectant-isopropyl liquid.**

| Isopropyl ratio in disinfectant (%) | ε' | ε" | Tan δ |
|---|---|---|---|
| 70 | 13.33 | 11.57 | 0.867 |
| 80 | 9.59 | 8.35 | 0.870 |
| 90 | 7.37 | 6.09 | 0.826 |
| 100 | 4.70 | 2.44 | 0.519 |

Fig 4 shows the measured dielectric spectra of the disinfectant-isopropyl sample in the frequency range from 1 to 5 GHz. Noteworthy, the value of dielectric parameter was found to be decreased exponentially with the increase of frequency for all the samples of various isopropyl contents. This is where the dielectric constant value was also seen to be decreased with the increase of isopropyl content. It can be concluded that the exponential decrease of ε with frequency is due to the total polarization drop resulting from a rapid change of dipole moment at higher frequencies (see Table 2 and Fig 4).

Fig 5 shows the photos of the PCB rectangular patch antenna which was fabricated by the LPKF Proto Mat E33 machine in the same dimensions and condition as of the simulation section. In the design of the sensor structure, the reflection coefficient (S11) parameter needs to resonate at a specified frequency. The value of this resonant frequency was seen to be $f$ = 4.656 GHz, as shown in Fig 6. However, this is not a commonly used frequency band, there are practical reasons to choose the proposed design and frequency range. For instance, it was aimed to avoid interference with environmental wireless frequencies and to compensate for the large difference of electrical properties of the sensor materials between 4–5 GHz.

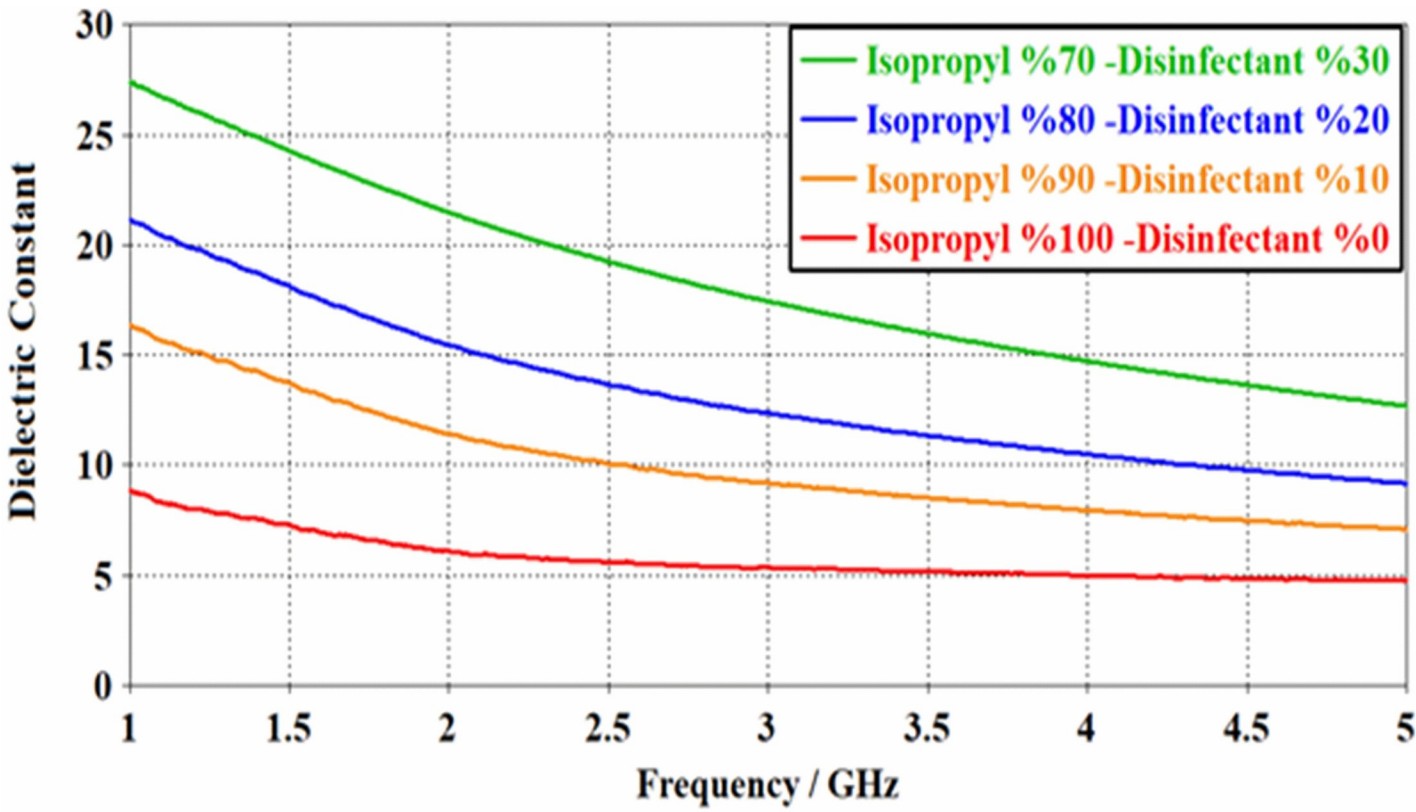

**Fig 4. Dielectric values for disinfectant-isopropyl in the frequency range from 1 to 5 GHz.**

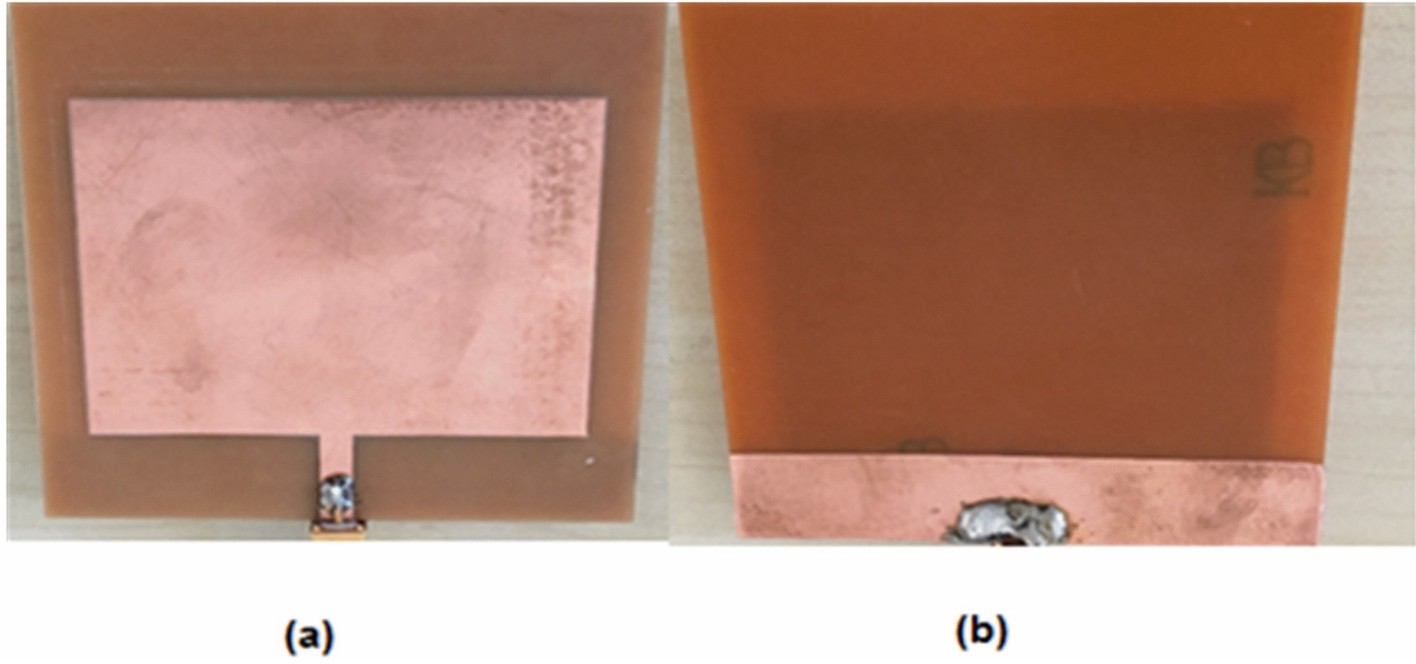

**Fig 5.** Photo of the fabricated PCB rectangular patch antenna: (a) frontside view and (b) backside view.

At the resonant frequency, the rectangular structure was optimized by means of geometric parameters study so that a best possible distant area is selected. Fig 6 shows that the patch antenna is resonating at the frequency of 4.656 GHz, where the $S_{11}$ value is approximately -12 dB. This indicated that the proposed antenna is able to produce an excellent propagation at the desired point. A such, the bandwidth of the proposed antenna was estimated to be 355 MHz at -10 dB level.

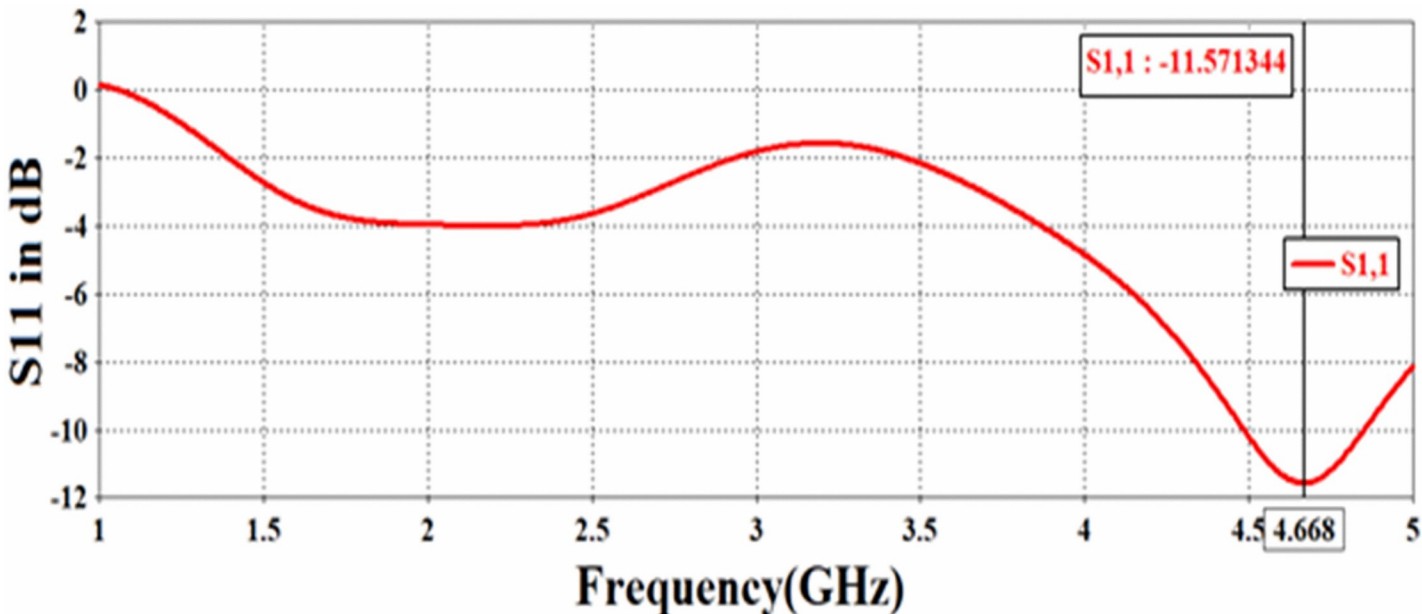

**Fig 6. Return loss ($S_{11}$) spectra of the proposed PCB rectangle patch antenna.**

**Table 3. Variations in resonant frequency, S₁₁ and dielectric parameters measured by the proposed antenna sensor with respect to different amounts of ethanol.**

| Ethanol content in wine (%) | Resonant Frequency (GHz) | S11 value (dB) | ε' | ε" | tan δ |
|---|---|---|---|---|---|
| 15 | 4.040 | -22.163 | 63.18 | 23.84 | 0.377 |
| 25 | 4.012 | -24.317 | 52.33 | 24.96 | 0.476 |
| 35 | 3.980 | -26.121 | 43.97 | 24.63 | 0.560 |
| 100 | 4.03 | -36.23 | 6.96 | 6.22 | 0.893 |

Table 3 shows the measurement results of the samples containing 15%, 25%,35% and 100% ethanol ratio in wine. It is worth mentioning that the ethanol content was increased by adding ethanol alcohol to the samples, starting with 15% ethanol in wine. The initial choice of 15% ethanol alcohol was made based on its presence as a control parameter in wine making. The resonant frequency at 15% was found to be 4.040 GHz. In actual application process, the effect of the ambient temperature on S11 values should be considered. When the content of ethanol was increased to 35%, which is our most concentrated sample, the resonant frequency was shifted to 3.980 GHz. Resonant frequency for pure ethanol was 4.03 GHz, whereas the reflection value in dB was found to be -22 dB -24 dB -26 dB and -36 dB for %15%25%35 and pure ethanol, respectively, which is higher than those reported before [18,12]. Taking a close look at the values in Table 3 one can find the presence of a significant linear shift in the resonant frequency with the increase of ethanol ratio As a result of this linearity, a total bandwidth of 60 MHz was achieved. The 60 MHz value allows us to easily and precisely estimate the intermediate values on the detection bandwidth.

It is worth noting that the resonant frequency is readily shifted with the change of ethanol content, as shown in from Fig 7. This unique response of the sensor can be interestingly utilized for the detection of various concentrations of ethanol in wine.

Fig 8 shows that both of the resonant frequency and dielectric constant is linearly decreased with the increase of ethanol ratio. The interval between the increments of 10% ethanol and the

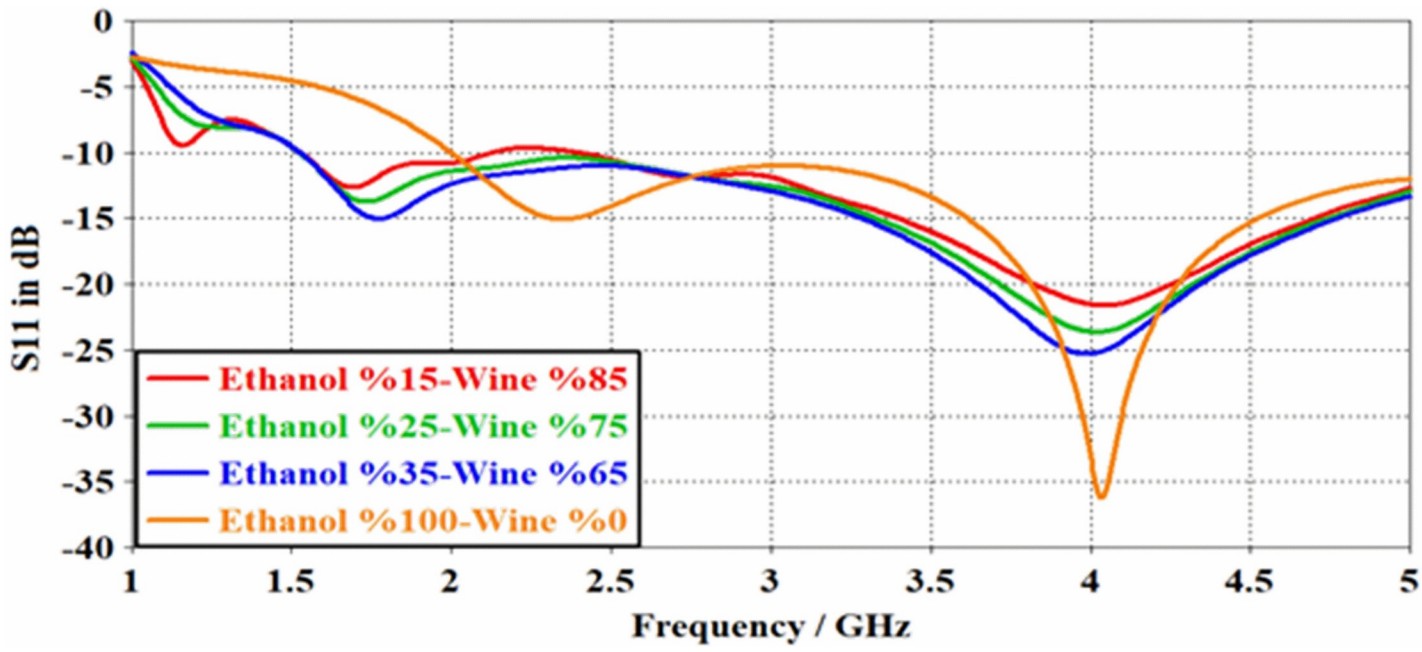

**Fig 7. Simulation results of S11 spectra achieved by PCB rectangular patch antenna for the wine-ethanol sample.**

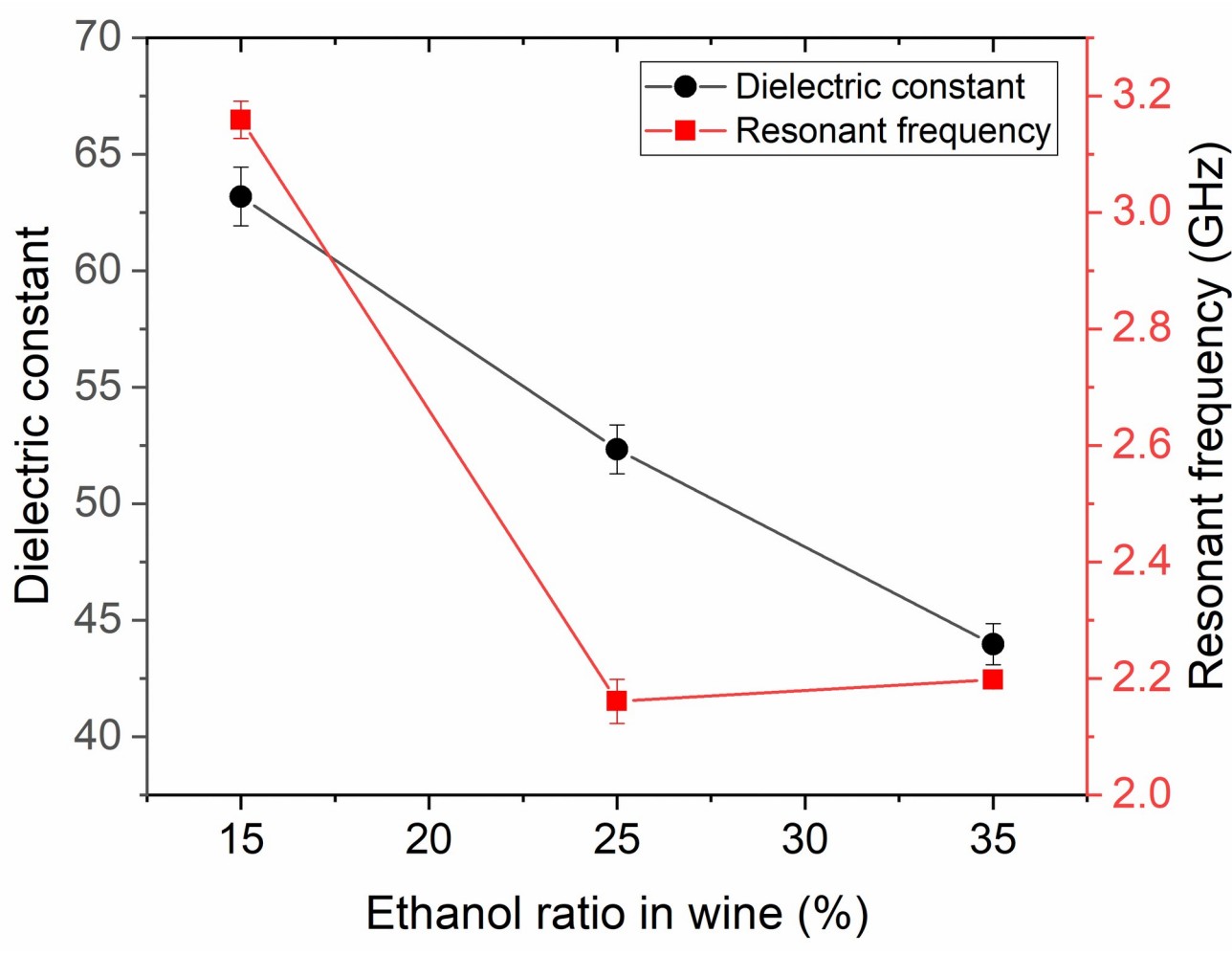

**Fig 8. Variation of the resonant frequency with ethanol ratio in wine for the PCB rectangular antenna sensor.**

changing structure shows that the intermediate values can be estimated. The changes shown by the curves in Fig 8 may indicate the amount of additive in a measured mixture, e.g. the amount of ethanol-alcohol in the wine. The electrical size, which varies according to the frequency, is small enough to detect these mixtures. This concluded that the proposed antenna-based sensor can be efficiently used for the detection of ethanol content in wine and other disinfectants.

Similarly, from the results shown in Fig 9 and Table 4, where the position of the resonant frequency was seen to be readily shifted with the change of isopropyl in disinfectant, a generalized conclusion of using the PCB sensor for the detection of various liquid samples can be drawn Measurements with hand disinfectant containing 70% isopropyl alcohol presented a resonant frequency of $f$ = 3.896 GHz, while this value was increased to $f$ = 4.18 GHz when the isopropyl content was raised to 100%. This is where the resonant frequencies for 80% and 90% isopropyl content were found to be 3.94 GHz and 4.024 GHz, respectively. According to these values, an approximately 288 MHz detection band can be obtained, which is superior than those reported by other researchers [14–16]. Besides, the reflection values in dB were seen to be -40.165 dB, -40.882 dB, -35.705 dB and -23.79 which are higher compared to those reported in literature [17–18].

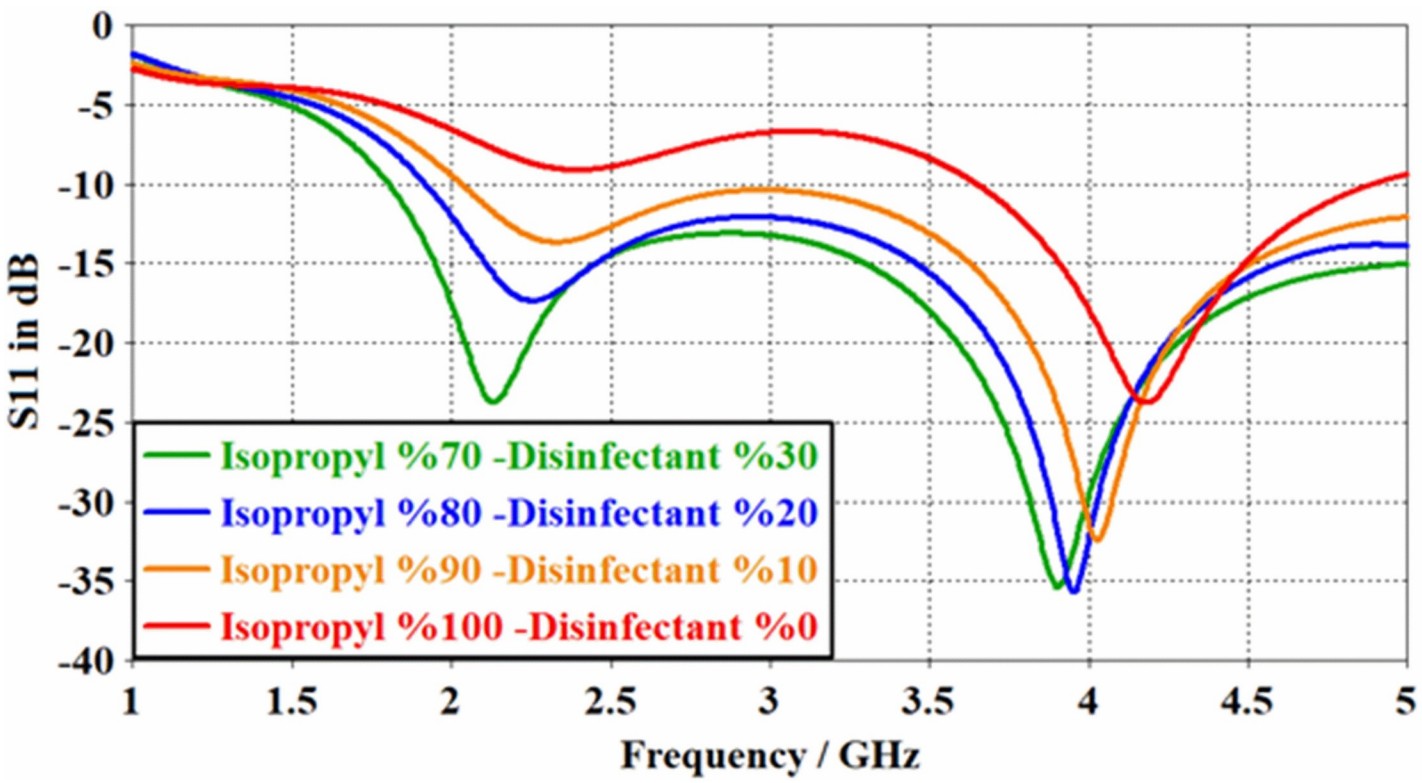

**Fig 9. Simulation results for the PCB rectangular patch antenna disinfectant-isopropyl sample.**

It is seen from Fig 10 that the change in resonant frequency with the isopropyl alcohol ratio can be directly correlated with the varied dielectric parameters. Variations shown by the curves of Fig 10 may indicate the amount of additive in a measured mixture, i.e. the amount of isopropyl alcohol in the disinfectant.

Figs 11 and 12 show the measured $S_{11}$ reflection coefficient parameter for the proposed PCB rectangular patch antenna and the resonant frequency shifts depending on the variation of the percentages of the ethanol alcohol in wine and isopropyl alcohol in disinfectant. The mixtures of the samples were prepared similar to that of the simulation, where 15%, 25% and 35% ethanol alcohol were added into the wine, separately. Besides, each of 70%, 80%, and 90% of isopropyl alcohol was mixed with disinfectant in order to have different types of liquid samples for the measurement purposes.

As it can be seen from Fig 11, there is a significant change in the resonant frequency when the ratio of ethanol alcohol is increased in the wine from 15% to 35% in steps of 10%. The resonant frequency was observed to be about 3.40 GHz, 3.60 GHz and 4.10 GHz for the aforementioned ethanol contents, respectively. The reflection values for the measurement result was

**Table 4. Resonant frequency, $S_{11}$ and dielectric for the isopropyl alcohol in disinfectant.**

| Isopropyl content in disinfectant (%) | Resonant Frequency (GHz) | $S_{11}$value (dB) | ε' | ε" | tan δ |
|---|---|---|---|---|---|
| 70 | 3.896 | -40.165 | 13.33 | 11.57 | 0.867 |
| 80 | 3.948 | -40.882 | 9.59 | 8.35 | 0.870 |
| 90 | 4.024 | -35.705 | 7.37 | 6.09 | 0.826 |
| 100 | 4.18 | -23.79 | 4.91 | 3.19 | 0.64 |

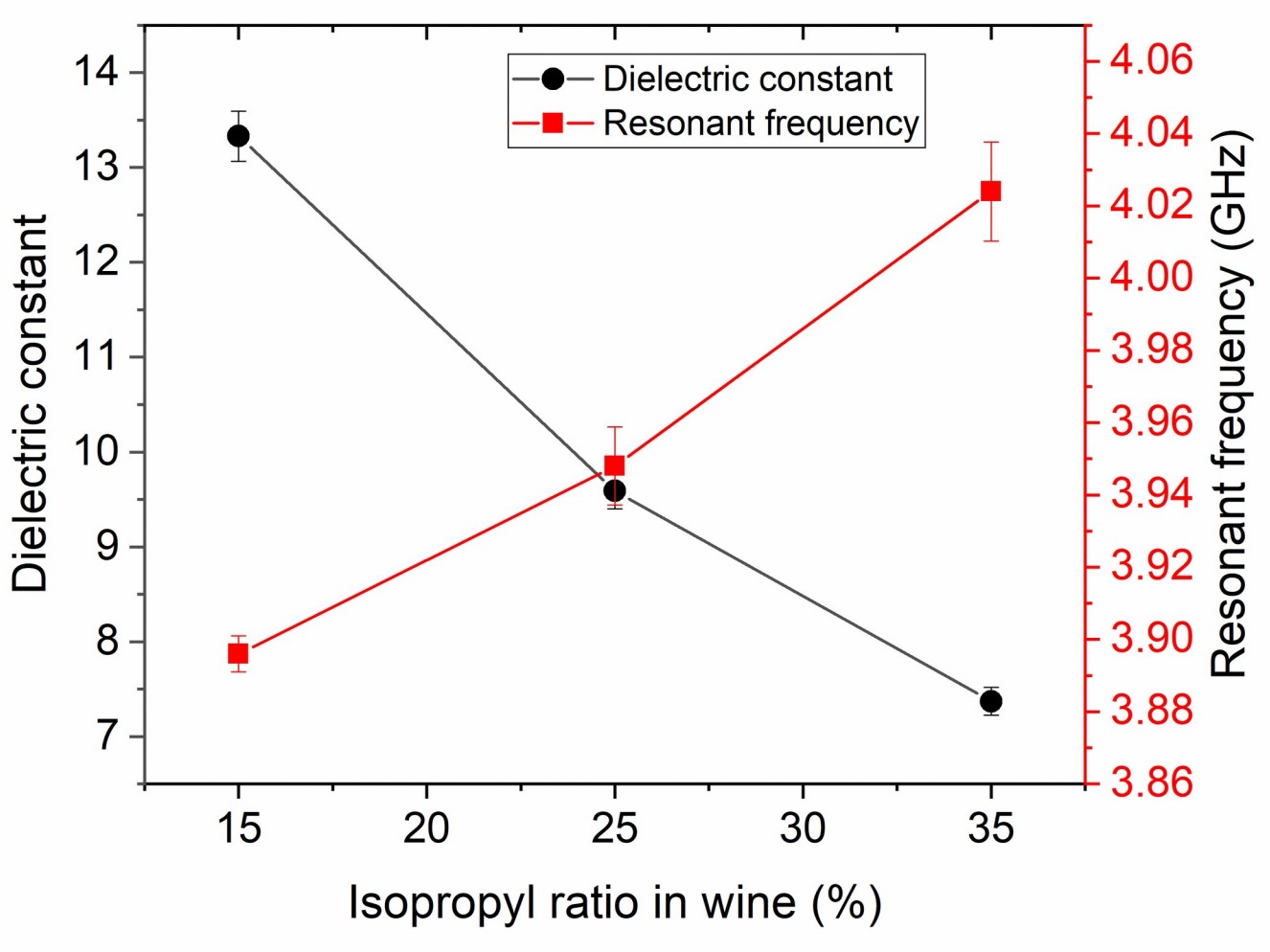

**Fig 10. Variations of the resonant frequency with isopropyl alcohol content detected by the proposed antenna sensor structure.**

obtained to be -18 dB, -25 dB and -43 dB for %15, %25 and %25 wine ethanol mixture samples, respectively. This was found to be in a good agreement with the simulation results, however a trivial deviation can be due to calibration error and manufacturing defects.

Fig 12 shows the measured $S_{11}$ parameter as a function of frequency, for the disinfectant sample with isopropyl concentrations of 70%, 80%, and 90% measured by the rectangular patch antenna. The resonant frequencies for the measured disinfectant with varied isopropyl were 3.78, 4.07, 4.17 and 4.3 GHz, respectively. This is equivalent to 3.86, 3.94, 4.02 and 4.18 GHz of the simulation results, which was seen to be better than those reported before [23, 24]. Noticeably, there is a 100 MHz difference between the measured and simulated results which might be due to the calibration error and manufacturing defects.

## Conclusion

A novel sensor based on printed circuit board (PCB) microstrip rectangular patch antenna was successfully fabricated and tested for the detection of various ratios of ethanol alcohol in wine and isopropyl alcohol in disinfectant. Results showed that any variations in the dielectric behavior of the liquid samples can be interestingly transduced to implement a useful linear

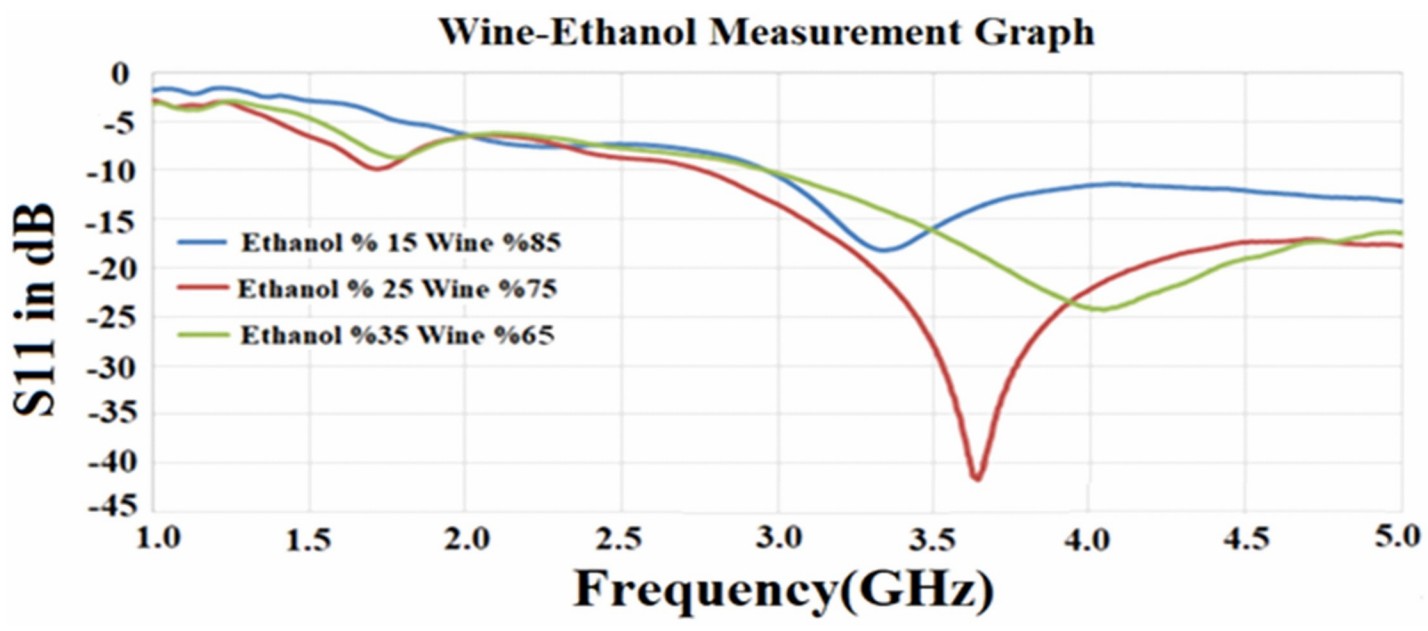

**Fig 11. Measured results for PCB rectangular patch antenna with wine-ethanol content.**

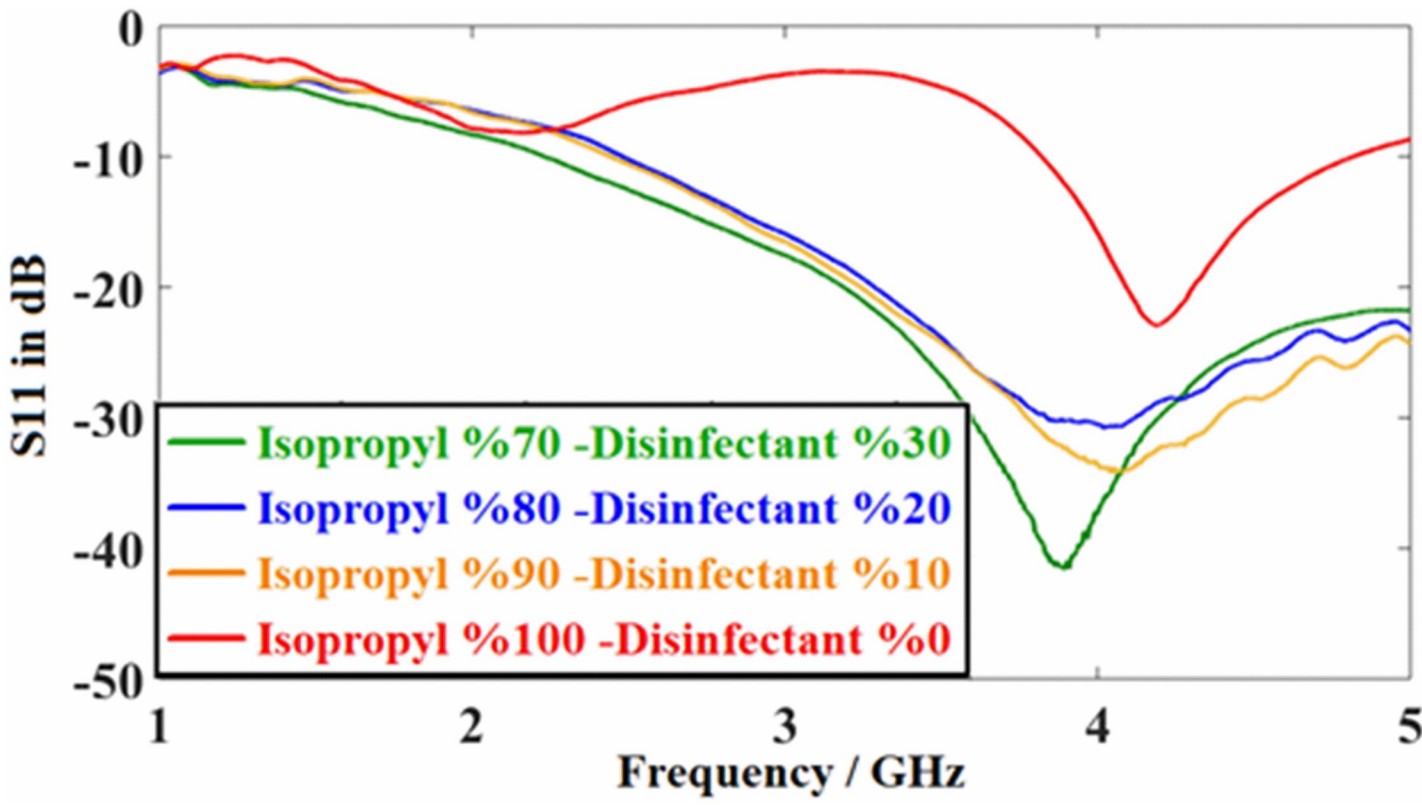

**Fig 12. Measured results for the antenna sensor structure with disinfectant-isopropyl content.**

shifting in the resonant frequency of the senor, through which the process of sensing liquid materials is realized. The sensor structure was found to have a low cost and high sensitivity, which can be readily utilized for multipurpose biological and chemical sensing applications.

## Author Contributions

**Funding acquisition:** Shengxiang Huang.

**Investigation:** Ayşegül Karatepe.

**Methodology:** Şekip Dalgac.

**Supervision:** Oğuzhan Akgöl, Lianwen Deng, Muharrem Karaaslan.

**Validation:** Emin Ünal.

**Writing – original draft:** Yadgar I. Abdulkarim.

**Writing – review & editing:** Fahmi F. Muhammadsharif, Halgurd N. Awl, Luo Heng.

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
