## [Decision Letter · Decision Letter 0]

19 Mar 2020

PONE-D-20-02183

Multi-Purpose Chemical Liquid Sensing Applications by Microwave Approach

PLOS ONE

Dear Dr huang,

Thank you for submitting your manuscript to PLOS ONE. After careful consideration, we feel that it has merit but does not fully meet PLOS ONE’s publication criteria as it currently stands. Therefore, we invite you to submit a revised version of the manuscript that addresses the points raised during the review process.

We would appreciate receiving your revised manuscript by May 03 2020 11:59PM. To enhance the reproducibility of your results, we recommend that if applicable you deposit your laboratory protocols in protocols.io, where a protocol can be assigned its own identifier (DOI) such that it can be cited independently in the future. For instructions see: http://journals.plos.org/plosone/s/submission-guidelines#loc-laboratory-protocols

We look forward to receiving your revised manuscript.

Kind regards,

Kalisadhan Mukherjee

Academic Editor

PLOS ONE

Additional Editor Comments (if provided):

The authors are suggested to carefully reply the comments/queries raised by the reviewers.

Journal Requirements:

"This work was partially supported by the National Key Research and Development Program of China (Grant no.2017YFA0204600), the National Natural Science Foundation of China (Grant no. 51802352) and the Fundamental Research Funds for the Central Universities of Central South University (Grant no.2018zzts355)."

"No"

5. Thank you for stating in your Acknowledgments Statement: "This work was partially supported by the National Key Research and Development Program of China (Grant no.2017YFA0204600), the National Natural Science Foundation of China (Grant no. 51802352) and the Fundamental Research Funds for the Central Universities of Central South University (Grant no.2018zzts355)."

6. Please upload a new copy of Figure 3-9 and 11-12 as the detail is not clear. Please follow the link for more information: http://blogs.PLOS.org/everyone/2011/05/10/how-to-check-your-manuscript-image-quality-in-editorial-manager/

Reviewers' comments:

Reviewer's Responses to Questions

**Comments to the Author**

1. Is the manuscript technically sound, and do the data support the conclusions?

Reviewer #1: Yes

Reviewer #2: Yes

2. Has the statistical analysis been performed appropriately and rigorously? 

Reviewer #1: N/A

Reviewer #2: Yes

3. Have the authors made all data underlying the findings in their manuscript fully available?

Reviewer #1: Yes

Reviewer #2: Yes

4. Is the manuscript presented in an intelligible fashion and written in standard English?

Reviewer #1: Yes

Reviewer #2: Yes

5. Review Comments to the Author

Reviewer #1: The study aims to propose a novel sensor structure with the goal to sense the ratio of ethanol alcohol in wine and isopropyl alcohol in disinfectant. The article needs to be revised, and here are my comments:

1. Abstract. The abstract needs to be rewritten taking into account the following points, although in accordance with the guidelines for authors: (i) issue addressed (why is there a need to propose a new sensor?); (ii) the aim of the study; (iii) brief information about methodology; (iv) main results and (v) final recommendations;

2. In general, the article needs to be revised to eliminate typing errors (i.e., Line 64 … “Three important…”,);

3. Novelties need to be better stressed, although the short literature review in the introduction is useful;

4. Current sections “Design of PCB Rectangular Patch Antenna” and “Measurement of Electrical Characteristics of the Liquid Samples” and in general the other sections of Materials and methods.

5. The paragraphs need to be simplified and summarised. It is necessary to eliminate the use of redundant sentences, in which the objectives/tasks of the instrument, slight references to literature, etc., are re-presented. The contents must be those indispensable to replicate the experiment, starting from the design of the sensor;

6. The results need to be discussed in more detail, including a comparison with the references mentioned in the introduction, which are assumed to be “overcome” with this contribution. This part is completely missing;

7. All figures need to be reworked with more professional graphics;

8. The conclusions need to be revised. First of all, any mention of existing work (at least formally) should be avoided. They must be synthesized (at least half of the current info) and above all authors must provide the home message of their research, suggesting fields of application.

Reviewer #2: The manuscript describes a novel design for PCB antenna for multi-purpose sensing applications. The topic is interesting and the quality of work carried out is appreciable. The work will be helpful in the advancement of the field. I would like to recommend the manuscript for publication. The following points may be noted before publication:

1. For the data represented in Fig. 7 and Fig. 9, error bars (standard deviation) may be used.

2. The conclusion section is too elaborate. The conclusion needs to be re-written in a concise manner.

6. PLOS authors have the option to publish the peer review history of their article (what does this mean?). If published, this will include your full peer review and any attached files.

Reviewer #1: No

Reviewer #2: No

---

## [Author Response · Author response to Decision Letter 0]

3 Apr 2020

Subject: Response to reviewers

Manuscript Number: PONE-D-20-02183

Manuscript Title: Multi-Purpose Chemical Liquid Sensing Applications by Microwave Approach

Prof. Kalisadhan Mukherjee

Academic Editor

PLOS ONE

 Thank you very much for your effort in handling our manuscript. We highly acknowledge the positive comments received from the reviewers and editor to improve the contents of our manuscript. The comments have helped us to further improve and strengthen our paper. The required revisions were performed and highlighted/track changed throughout the manuscript. Please find below our response to the reviewers’ comments accordingly.

REVİEWER #1 

 1.Abstract. The abstract needs to be rewritten taking into account the following points, although in accordance with the guidelines for authors: (i) issue addressed (why is there a need to propose a new sensor?); (ii) the aim of the study; (iii) brief information about methodology; (iv) main results and (v) final recommendations;

Thank you for your nice recommendation. We have revised the abstract as below:

In this work, a novel sensor based on printed circuit board (PCB) microstrip rectangular patch antenna is proposed to detect different ratios of ethanol alcohol in wines and isopropyl alcohol in disinfectants. The proposed sensor was designed by finite integration technique (FIT) based high-frequency electromagnetic solver (CST) and was fabricated by Proto Mat E33 machine. To implement the numerical investigations, dielectric properties of the samples were first measured by a dielectric probe kit then uploaded into the simulation program. Results showed a linear shifting in the resonant frequency of the sensor when the dielectric constant of the samples were changed due to different concentrations of ethanol alcohol and isopropyl alcohol. A good agreement was observed between the calculated and measured results, emphasizing the usability of dielectric behavior as an input sensing agent. It was concluded that the proposed sensor is viable for multipurpose chemical sensing applications. 

2.In general, the article needs to be revised to eliminate typing errors (i.e., Line 64 … “Three important…”,)

Thank you for the valuable comment.

The revised manuscript was thoroughly proofread by an English language professional to avoid possible typos and grammatical errors.

3.Novelties need to be better stressed, although the short literature review in the introduction is useful;

Thank you for the valuable comment. A clear expression to the novelty of the work was provided, as stated below:

The novelty of this work is to suggest a simple and cheap patch antenna sensor which can be readily utilized for the determination of ethanol ratio in wine and disinfectant. Additionally, the new proposed designs can be used to develop a portable sensor instrument to be easily used for multipurpose sensing applications. 

4.Current sections “Design of PCB Rectangular Patch Antenna” and “Measurement of Electrical Characteristics of the Liquid Samples” and in general the other sections of Materials and methods.

Thank you for the valuable comment. The headings of the sections and subsections in the manuscript were also revised. 

5.The paragraphs need to be simplified and summarised. It is necessary to eliminate the use of redundant sentences, in which the objectives/tasks of the instrument, slight references to literature, etc., are re-presented. The contents must be those indispensable to replicate the experiment, starting from the design of the sensor

Thank you for your nice comment. We have made our best effort to present the paragraphs in a more clear and understandable fashion. The unnecessary writings were eliminated in the way that they do not affect the intended meaning of the contents.

6.The results need to be discussed in more detail, including a comparison with the references mentioned in the introduction, which are assumed to be “overcome” with this contribution. This part is completely missing

Thank you for your nice comment. Based on this comment, further elaboration on the obtained results and their comparison to those reported in literature was added into the revised version of the manuscript, as below:

Resonant frequency for pure ethanol was 4.03 GHz, whereas the reflection value in dB was found to be -22 dB -24 dB -26 dB and -36 dB for %15 %25 %35 and pure ethanol, respectively, which is higher than those reported before [18,12]. Taking a close look at the values in Table 3 one can find the presence of a significant linear shift in the resonant frequency with the increase of ethanol ratio As a result of this linearity, a total bandwidth of 60 MHz was achieved. The 60 MHz value allows us to easily and precisely estimate the intermediate values on the detection bandwidth.

Measurements with hand disinfectant containing 70% isopropyl alcohol presented a resonant frequency of f = 3.896 GHz, while this value was increased to f = 4.18 GHz when the isopropyl content was raised to 100 %. This is where the resonant frequencies for 80% and 90% isopropyl content were found to be 3.94 GHz and 4.024 GHz, respectively. According to these values, an approximately 288 MHz detection band can be obtained, which is superior than those reported by other researchers [14-16]. Besides, the reflection values in dB were seen to be -40.165 dB, -40.882 dB, -35.705 dB and -23.79 which are higher compared to those reported in literature [17-18].

[12] Liu, W., Sun, H., & Xu, L. (2018). A microwave method for dielectric characterization measurement of small liquids using a metamaterial-based sensor. Sensors, 18(5), 1438.

[14] Altintas, O., Aksoy, M., Akgol, O., Unal, E., Karaaslan, M., & Sabah, C. (2017). Fluid, strain and rotation sensing applications by using metamaterial based sensor. Journal of The Electrochemical Society, 164(12), B567-B573.

[16] Ebrahimi, A., Withayachumnankul, W., Al-Sarawi, S., & Abbott, D. (2013). High-sensitivity metamaterial-inspired sensor for microfluidic dielectric characterization. IEEE Sensors Journal, 14(5), 1345-1351.

[17] Bakir, M. (2017). Electromagnetic-based microfluidic sensor applications. Journal of the electrochemical society, 164(9), B488-B494.

 [18] Ling, K., Yoo, M., Su, W., Kim, K., Cook, B., Tentzeris, M. M., & Lim, S. (2015). Microfluidic tunable inkjet-printed metamaterial absorber on paper. Optics express, 23(1), 110-120.

7.All figures need to be reworked with more professional graphics

The figures were regenerated to enhance the quality of their appearance, as shown below.

Figure 5. PCB rectangle patch antenna return loss graph (S11)

Figure 7. Variation of the resonance frequency with ethanol ratio in wine for the PCB rectangular antenna sensor

Figure 9.Variations of the resonance frequency with isopropyl alcohol content detected by the proposed antenna sensor structure.

Figure 11. Measured results for PCB rectangular patch antenna with wine-ethanol content 

8.The conclusions need to be revised. First of all, any mention of existing work (at least formally) should be avoided. They must be synthesized (at least half of the current info) and above all authors must provide the home message of their research, suggesting fields of application.

Thank you for your valuable comment. The conclusion part revised, as stated below:

A novel sensor based on printed circuit board (PCB) microstrip rectangular patch antenna was successfully fabricated and tested for the detection of various ratios of ethanol alcohol in wine and isopropyl alcohol in disinfectant. Results showed that any variations in the dielectric behavior of the liquid samples can be interestingly transduced to implement a useful linear shifting in the resonant frequency of the senor, through which the process of sensing liquid materials is realized. The sensor structure was found to have a low cost and high sensitivity, which can be readily utilized for multipurpose biological and chemical sensing applications. 

REVİEWER 2

1.For the data represented in Fig. 7 and Fig. 9, error bars (standard deviation) may be used

Thank you for the valuable comment. We have included the error bar to the graphs based on the standard deviation data.

Figure 8. Variation of the resonance frequency with ethanol ratio in wine for the PCB rectangular antenna sensor

Figure 10.Variations of the resonance frequency with isopropyl alcohol content detected by the proposed antenna sensor structure.

The conclusion section is too elaborate. The conclusion needs to be re-written in a concise manner.

Thank you for your valuable comment. The conclusion part was shortened as given below;

In summary, a novel design for PCB rectangular patch antenna was presented for sensing application at 1-5 GHz frequency range. According to the numerical results, the proposed sensor structure was successfully designed for the real-time, fast and accurate detection of ethanol and isopropyl alcohol as a biochemical sensor in the wine-ethanol and disinfectant-isopropyl mixtures. It was observed that the resonance frequency changed linearly according to the samples formed by increasing the 10% ethanol alcohol and isopropyl alcohol. As a result of this linearity, it is seen that there is a detection bandwidth of 60 MHz and 128 MHz which means that 30 Mhz and 64 MHz resonant frequecny shifting observed for %10 step of chancing in ethanol and isopropyl alcohol. To support the numerical analysis and compare the results, we fabricated the proposed sensor structure. Practically obtained results were seen to be in a good agreement with the simulation ones. The determination of ethanol content even at larger resolution is critic to reduce selling of unrecorded wine and disinfectant which results in several health problems including death. That’s why the proposed sensor structure can be used in biosensing and chemical sensing applications in order to avoid death tools and measuring ethanol content in wine and disinfectant mixture. For future studies, it is aim to take intermediate value of ethanol content enhanced up to 2% ratio and design an microwave circuit to adapt it portable design.

---

## [Decision Letter · Decision Letter 1]

16 Apr 2020

Multipurpose chemical liquid sensing applications by microwave approach

PONE-D-20-02183R1

Dear Dr. huang,

We are pleased to inform you that your manuscript has been judged scientifically suitable for publication and will be formally accepted for publication once it complies with all outstanding technical requirements.

With kind regards,

Kalisadhan Mukherjee

Academic Editor

PLOS ONE

Additional Editor Comments (optional):

The reviewers have recommended the acceptance of the manuscript. It can now be accepted for publication.

Reviewers' comments:

Reviewer's Responses to Questions

**Comments to the Author**

1. If the authors have adequately addressed your comments raised in a previous round of review and you feel that this manuscript is now acceptable for publication, you may indicate that here to bypass the “Comments to the Author” section, enter your conflict of interest statement in the “Confidential to Editor” section, and submit your "Accept" recommendation.

Reviewer #1: All comments have been addressed

Reviewer #2: All comments have been addressed

2. Is the manuscript technically sound, and do the data support the conclusions?

Reviewer #1: Yes

Reviewer #2: Yes

3. Has the statistical analysis been performed appropriately and rigorously? 

Reviewer #1: N/A

Reviewer #2: Yes

4. Have the authors made all data underlying the findings in their manuscript fully available?

Reviewer #1: Yes

Reviewer #2: Yes

5. Is the manuscript presented in an intelligible fashion and written in standard English?

Reviewer #1: Yes

Reviewer #2: Yes

6. Review Comments to the Author

Reviewer #1: The manuscript has been improved by accepting all comments.

In my opinion the contribution is more suitable to be published on a more sectoral section of the website.

Reviewer #2: The authors have addressed my comments and I am satisfied now. I recommend the manuscript for publication.

7. PLOS authors have the option to publish the peer review history of their article (what does this mean?). If published, this will include your full peer review and any attached files.

Reviewer #1: No

Reviewer #2: No

---

## [Editor Report · Acceptance letter]

22 Apr 2020

PONE-D-20-02183R1 

Multipurpose chemical liquid sensing applications by microwave approach 

Dear Dr. Huang:

I am pleased to inform you that your manuscript has been deemed suitable for publication in PLOS ONE. Congratulations! Your manuscript is now with our production department. 

With kind regards,

on behalf of

Dr. Kalisadhan Mukherjee 

Academic Editor

PLOS ONE